# Performance of Platelet Counting in Thrombocytopenic Samples: Comparison between Mindray BC-6800Plus and Sysmex XN-9000

**DOI:** 10.3390/diagnostics12010068

**Published:** 2021-12-29

**Authors:** Hanah Kim, Mina Hur, Gun-Hyuk Lee, Seung-Wan Kim, Hee-Won Moon, Yeo-Min Yun

**Affiliations:** Department of Laboratory Medicine, Konkuk University School of Medicine, Seoul 05030, Korea; md.hkim@gmail.com (H.K.); leegunhyuk93@gmail.com (G.-H.L.); kswsun@kuh.ac.kr (S.-W.K.); hannasis@hanmail.net (H.-W.M.); ymyun@kuh.ac.kr (Y.-M.Y.)

**Keywords:** platelet, thrombocytopenia, precision, Mindray BC-6800Plus, Sysmex XN-9000

## Abstract

The performance of platelet (PLT) counting in thrombocytopenic samples is crucial for transfusion decisions. We compared PLT counting and its reproducibility between Mindray BC-6800Plus (BC-6800P, Mindray, Shenzhen, China) and Sysmex XN-9000 (XN, Sysmex, Kobe, Japan), especially focused on thrombocytopenic samples. We analyzed the correlation and agreement of PLT-I channels in both analyzers and BC-6800P PLT-O mode and XN PLT-F channel in 516 samples regarding PLT counts. Ten thrombocytopenic samples (≤2.0 × 10^9^/L by XN PLT-F) were measured 10 times to investigate the reproducibility with the desirable precision criterion, 7.6%. The correlation of BC-6800P PLT-I and XN PLT-I was arranged moderate to very high; but the correlation of BC-6800P PLT-O and XN PLT-F was arranged high to very high. Both BC-6800P PLT-I vs. XN PLT-I and BC-6800P PLT-O vs. XN PLT-F showed very good agreement (κ = 0.93 and κ = 0.94). In 41 discordant samples between BC-6800P PLT-O and XN PLT-F at transfusion thresholds, BC-6800P PLT-O showed higher PLT counts than XN-PLT-F, except the one case. BC-6800P PLT-O exceeded the precision criterion in one of 10 samples; but XN PLT-F exceeded it in six of 10 samples. BC-6800P would be a reliable option for PLT counting in thrombocytopenic samples with good reproducibility.

## 1. Introduction

Clinical practice guidelines and researches have suggested that the threshold for prophylactic platelet (PLT) transfusion can be safely set at 20 × 10^9^/L, 10 × 10^9^/L, or even lower according to the patients’ condition and the reversibility of their bone marrow failure [1,2,3,4,5,6,7]. The performance of PLT counting in thrombocytopenic samples is crucial for clinical practice, especially in PLT transfusion management [8].

The international reference method is based on flow cytometry using monoclonal antibodies (i.e., CD61 and CD41) for PLT counting proposed by the International Council for Standardization in Haematology (ICSH) [9,10]. Flow cytometric analysis, however, has some obstacles, such as lack of widespread availability of dedicated instrumentation, standardization issues, labor intensiveness, and the higher cost of its analysis compared with modern automated hematology analyzers using impedance and/or optic detection with fluorescent labeling and flow cytometry [8,9,10,11]. Mindray BC-6800Plus (BC-6800P, Mindray Bio-Medical Electronics Co., Ltd., Shenzhen, China) uses an impedance channel (PLT-I) and an optic detection with fluorescent labeling mode (PLT-O). Sysmex XN-9000 (XN, Sysmex, Kobe, Japan) uses an impedance channel (PLT-I), optical channels (PLT-O), and fluorescent channel (PLT-F), separately.

Even though state-of-the-art technologies have been implemented in automated hematology analyzers and clinical laboratories, standardization and harmonization of PLT counts and PLT indices have not been achieved yet, especially in thrombocytopenic samples [8,12,13]. The UK National External Quality Assessment Scheme for General Haematology reported that the imprecision increased at lower levels of PLT counts, showing a mean coefficient of variance (CV) as high as 15–32% at transfusion thresholds [14,15,16]. Their imprecisions showed still higher values than biological variability of PLT, approximately 4–7% [15,16,17].

Analytical performance specifications of PLT counting and PLT parameters (MPV, PCT, PDW, and P-LCR) on XN were determined using 43 normal samples, and the overall performance was acceptable with at least a desirable reproducibility [18]. Differently from BC-6800, BC-6800P measures PLT counts using multi-fold counting (×8 times) automatically in thrombocytopenic samples [19,20]. BC-6800P showed acceptable precision using six thrombocytopenic samples (<20 × 10^9^/L) [20]. Within-run precision of PLT-I and PLT-O on BC-6800P, and PLT-I, PLT-O, and PLT-F on XN were determined in 10 replicates using four thrombocytopenic samples (10, 20, 40, and 50 × 10^9^/L) [21]. However, these studies have been conducted with a limited number of thrombocytopenic samples to compare PLT counts in pairs by PLT counting methods and to evaluate the reproducibility of PLT counting. Therefore, further studies are necessary with an increased number of thrombocytopenic samples especially focused on transfusion thresholds.

We evaluated the performance of PLT counting of BC-6800P compared with XN, which is currently used as a routine hematology analyzer in our laboratory, according to their PLT counting methods. Furthermore, we focused on the reproducibility of PLT counting at PLT transfusion thresholds. Moreover, we compared five clinically reportable PLT parameters available on the BC-6800P with those of XN.

## 2. Materials and Methods

### 2.1. Study Population and Design

A total of 516 blood samples were collected consecutively from the individuals who visited the Konkuk University Medical Center (KUMC) from April to June 2019. The study protocol was designed following the criteria of the Declaration of Helsinki and approved by the Institutional Review Board of KUMC (KUH1200093). Because this study was conducted using remnant blood samples from the study population and the PLT counts driven from this study were not used for the transfusion decision, getting written informed consent from the enrolled patients was waived.

The peripheral blood samples (3 mL) were collected directly into VACUETTE EDTA K3 tubes (Greiner Bio-One, Kremsmünster, Austria), and complete blood counts (CBC) was analyzed within two hours using both BC-6800P and XN. PLT counts measured by XN PLT-F were considered a reference method, and PLT counts measured by BC-6800P were considered a comparative method. All samples were divided into five groups according to the PLT counts; ≤10 × 10^9^/L (*n* = 38), 11–20 × 10^9^/L (*n* = 112), 21–50 × 10^9^/L (*n* = 111), 51–100 × 10^9^/L (*n* = 60), and >100 × 10^9^/L (*n* = 195). To further investigate the impact of PLT counting at PLT transfusion thresholds, the concordance of PLT counts on both analyzers were evaluated for transfusion thresholds of 10 × 10^9^/L, 20 × 10^9^/L, and 50 × 10^9^/L. The precision of PLT counts and clinically reportable PLT parameters in 10 thrombocytopenic (3.9–18.5 × 10^9^/L by XN PLT-F) samples on BC-6800P and XN was evaluated. Each of these samples was measured separately 10 consecutive times within two hours.

### 2.2. Assays

Compared with its previous version (BC-6800), the BC-6800P has adopted a new fluorescent dye (fluorescent retic (FR) dye) and a new erythrocyte-reticulocyte-platelet (ERP) channel technologies for PLT counting (PLT-O) and PLT parameters. The ERP channel does not lyse RBCs but spherizes RBCs using the DR diluent to avoid microcytic and fragmented RBCs interference. In the ERP channel, the RNA content of red blood cells (RBCs) and PLTs is stained by a new FR dye, which indicates asymmetric cyanine dye in the solution. Using FR dye, the reticulocytes and PLTs are more specifically stained with stronger fluorescent signals that bring more accurate results. With this technology, the BC6800P can measure smaller size PLTs (≥2 fL), enabling multi-fold counting (×8 times) at a low level of PLT counts. In addition to PLT counts, the BC-6800P reports 10 PLT parameters, including six clinically reportable parameters (%-IPF, MPV, PCT, PDW, P-LCC, and P-LCR) and four research-use-only parameters (A-IPF, H-IPF, MPC, and MPM) [19,22].

The XN PLT-F adopted fluorescence labelling flow cytometry as well as 5-fold samples volume compared to the XN PLT-I, which uses impedance measurement using hydrodynamic focusing. In addition to PLT counts, the XN reports seven PLT parameters, including six clinically reportable parameters (%-IPF, A-IPF, MPV, PCT, PDW, and P-LCR) and one research-use-only parameter (H-IPF) [18].

In this study, PLT counts and clinically reportable PLT parameters were measured in the ERP channel, and those values were compared with PLT counts and clinically reportable PLT parameters (%-IPF, MPV, PCT, PDW, and P-LCR) of XN.

For the study period, three levels of quality control materials in liquid provided by the manufacturers were run daily on BC-6800P and XN. The mean within-laboratory precision was <4.0% on both analyzers. The linearity range of PLT counts was 0–1000 × 10^9^/L, and the carryover was ≤1.0% on both analyzers. The i-Message value on BC-6800P, which provides more quantitative and comprehensive information on the severity of the abnormality for PLT clumping flags in each sample, was checked to confirm the absence of PLT clumping in each sample; its reportable range is 1–100 (cut-off value, 40), and no sample showed a PLT clumping flag (range, 1–26). For the XN, the probability of finding PLT clumping is expressed by Q value, which provides the degree of positive or negative results of the flag on a scale of 0 to 300 (cut off value is ≥100). The PLT clumping flag was also not observed on XN.

### 2.3. Statistical Analysis

Grubb’s test was performed for outlier detection, and all the measurements showed no apparent outliers or a statistical outlier. The Passing–Bablok regression and the Bland–Altman plot were analyzed in five groups separately for comparison of two analyzers. In the Passing–Bablok regression, the correlation coefficient (r) was interpreted as follows: ≤0.30, negligible; 0.30–0.50, low; 0.50–0.70, moderate; 0.70–0.90, high; and ≤0.90, very high [23]. On the Bland–Altman plot, the mean differences and limit of agreement (LOA) were interpreted informally to visualize how big the mean discrepancy is. Agreement was expressed using the Cohen’s weighted kappa (κ) showing linear weights as follows: <0.2 poor, 0.21–0.40 fair, 0.41–0.60 moderate, 0.61–0.80 good, and 0.81–1.0 very good [24]. The precision was expressed as the percentage CV (CV%), and less than 7.6 CV% was considered as a desirable specification of precision following the desirable specification of precision of the European Federation of Clinical Chemistry and Laboratory Medicine (EFLM) [25]. MedCalc Statistical Software (version 20.0.4, MedCalc Software Ltd., Ostend, Belgium) was used. *p*-values less than 0.05 were considered statistically significant.

## 3. Results

In the comparison of PLT counts between BC-6800P PLT-I vs. XN PLT-I and BC-6800P PLT-O vs. XN PLT-F in five groups of PLT counts, the overall correlation of BC-6800P PLT-I vs. XN PLT-I and BC-6800P PLT-O vs. XN PLT-F were very high (r = 0.99 and r = 1.00). However, in thrombocytopenic samples (≤100 × 10^9^/L), the correlation of BC-6800P PLT-I vs. XN PLT-I was moderate to high (0.57–0.94). The correlation of BC-6800P PLT-O vs. XN PLT-F was high to very high (0.76–0.94). The mean differences of BC-6800P PLT-I vs. XN PLT-I were broader than BC-6800P PLT-O vs. XN PLT-F (−7.7–3.6 vs. −2.5–3.2). In samples with PLT counts less than 100 × 10^9^/L, BC-6800P PLT-O vs. XN PLT-F showed a negative trend in PLT counts (Table 1).

Both BC-6800P PLT-I vs. XN PLT-I and BC-6800P PLT-O vs. XN PLT-F showed very good agreement (κ = 0.93 and κ = 0.94). In thrombocytopenic samples (≤50 × 10^9^/L, *n* = 261), both BC-6800P PLT-I vs. XN PLT-I and BC-6800P PLT-O vs. XN PLT-F showed good agreement (κ = 0.76 and κ = 0.78). We noted 41 discordant samples between BC-6800P PLT-O vs. XN PLT-F at transfusion thresholds (10 × 10^9^/L and 20 × 10^9^/L) (Table 2). In those samples, BC-6800P PLT-O showed higher PLT counts than XN-PLT-F, except the one case (case #41) (14 × 10^9^/L vs. 23 × 10^9^/L) (Figure 1).

In 10 thrombocytopenic samples (≤20.0 × 10^9^/L by XN PLT-F), BC-6800P PLT-I exceeded the desirable specification of precision of the EFLM biological variation 7.6% in eight of 10 samples (range, 8.1–32.9%). However, the precision of XN PLT-I exceeded the criterion in all 10 thrombocytopenic samples (range, 8.1–27.9%). The precision of BC-6800P PLT-O exceeded the criterion only in one of 10 thrombocytopenic samples (9.5%) However, the precision of XN PLT-F exceeded 7.6% in six of 10 thrombocytopenic samples (8.1–21.7%) (Table 3). Regarding clinically reportable PLT parameters, both BC-6800P and XN reported IPF values in all 10 thrombocytopenic samples. The precision of BC-6800P IPF was narrower than those of XN IPF (5.5–25.5% vs. 9.0–46.1%). For MPV, PCT, PDW, and P-LCR, BC-6800P reported values in all 10 thrombocytopenic samples. However, XN did not report values in four of 10 cases (Table 4).

## 4. Discussion

This is the first study that evaluated the performance of PLT counting of BC-6800P compared with XN in thrombocytopenic samples, in particular, focused on reproducibility at the PLT transfusion decision threshold. Furthermore, we explored clinically reportable PLT parameters in thrombocytopenic samples. The fluorescence methods (BC-6800P PLT-O and XN PLT-F) showed better mean differences, correlation, and agreement than impedance methods (BC-6800P PLT-I and XN PLT-I), especially in thrombocytopenic samples less than or equal to 20 × 10^9^/L (Table 1 and Table 2). It is well known that the fluorescence methods have more advantages over impedance methods, especially for samples with low PLT counts [11,12,20,21,26]. Our data support the previous findings. Due to extrapolating and calculating the PLT count in the area within a specific size range (approximately between 20 fL and 60 fL) of PLT volume histogram in the impedance method, inaccuracies of PLT count rely on cell size and on calculating logics. The French-speaking Cellular Hematology Group commented that simultaneous measurement of fluorescence and scattered light gives a more accurate PLT counting, particularly when large PLT has to be distinguished from other relatively large particles, such as RBC fragments [27].

PLT counting methods seem to affect the precision, especially in thrombocytopenic samples (Table 3). The fluorescence methods of both analyzers showed better reproducibility than impedance methods. In addition, BC-6800P PLT-O showed better reproducibility than XN PLT-F. This finding is in line with a previous finding from Gioia M et al. [21]. They reported that CV increased as the PLT counts decreased in all nine different hematology analyzers. In that study, BC-6800P PLT-O and XN-20 PLT-F showed acceptable precision in thrombocytopenic samples of less than or equal to 10 × 10^9^/L. However, BC-6800P PLT-I and XN PLT-I did not show the acceptable precision.

Clinically reportable PLT parameters, including IPF, are useful for diagnosing and monitoring thrombocytopenia. Based on the present and previous findings, the PLT counting methods also seem to affect precision and reference intervals of clinically reportable PLT parameters (Table 4) [19,21,28,29,30]. Thus, further exploration on clinically reportable PLT parameters across PLT counting methods is necessary to promote their clinical utilization.

EFLM established a biological variation database by systematic literature meta-analysis [25]. A 7.6 CV% was a median CV estimate of the within-subject of EFLM and considered a desirable specification of precision for PLT counts. However, as EFLM mentioned, the meta-analysis for estimates came from healthy populations [25]. In thrombocytopenic conditions, megakaryopoiesis and within-subject CV could differ from a healthy individual. Therefore, EFLM has a plan to develop a meta-analysis from different study groups, including disease settings. In line with previous studies, this study showed that both BC-6800P and XN did not satisfy the previously reported acceptable precision for analytical performance specification in thrombocytopenic samples [18,21,25]. Nevertheless, BC-6800P showed narrower ranges of CV for the parameters than XN [21]. Thus, further investigations are necessary to develop biological variations under various conditions.

BC-6800P PLT-O enumerated PLT counts higher and more precisely than XN PLT-F in thrombocytopenic samples (≤100 × 10^9^/L) (Table 1, Table 2 and Table 3 and Figure 1). Especially in 41 discordant samples at transfusion thresholds (10 × 10^9^/L and 20 × 10^9^/L), BC-6800P PLT-O showed higher PLT counts than XN-PLT-F, except the one case (Figure 1). Considering better reproducibility of BC-6800P, it has the strength for reducing unnecessary transfusion management.

This study has several limitations. First, we did not confirm the PLT counts using immunological PLT counting method that is endorsed by ICSH and ISLH [9,10]. Instead of confirming PLT counts using the reference method, we evaluated the reproducibility of PLT counting by replicate measurements under similar conditions. Second, we checked PLT clumping flag and the i-Message value for PLT clumping flag to exclude pseudothrombocytopenia. However, we could not verify the performance of the PLT clumping flag and i-Message value on a peripheral blood smear, which is considered a reference method. We could not fully follow the Clinical and Laboratory Standards Institute guidelines EP05-A3 to evaluate the precision [31]. The guidelines are assumed to be stable with no degradation of samples during the data collection period and mentioned its limitation for the samples with inadequate stability, such as RBC counts and blood gas determinations. We explored the precision of clinically reportable PLT parameters in thrombocytopenic samples. However, these results could not cover the full range of innovative PLT parameters in various clinical situations. Further investigations should include a sufficient number of cases.

This is the first study exploring the comparability and reproducibility of PLT counting in thrombocytopenic samples on BC-6800P and XN, including clinically reportable PLT parameters. The BC-6800P would be a promising and reliable option in clinical hematology laboratories and PLT transfusion decisions with good reproducibility in thrombocytopenic samples.

## Figures and Tables

**Figure 1 diagnostics-12-00068-f001:**
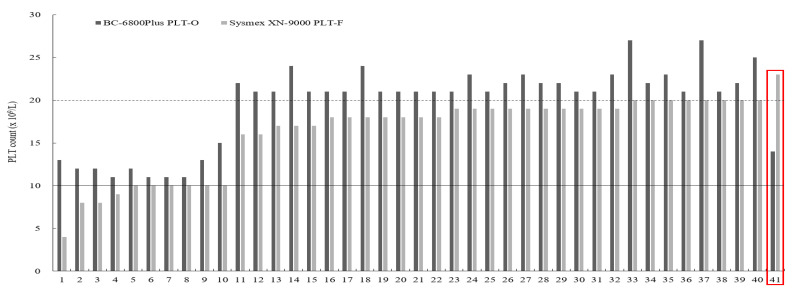
Discordant platelet (PLT) counts at transfusion threshold (10 × 10^9^/L and 20 × 10^9^/L) between BC-6800Plus PLT-O and Sysmex XN-9000 PLT-F (*n* = 41). Dark grey indicates PLT counts of BC-6800Plus PLT-O, and light grey indicates PLT counts of Sysmex XN-9000 PLT-F. The solid line indicates 10 × 10^9^/L, and the dashed line indicates 20 × 10^9^/L. Except case 41 (indicted by red rectangle), PLT counts of BC-6800Plus were higher than PLT counts of Sysmex XN-9000 PLT-F.

**Table 1 diagnostics-12-00068-t001:** Correlation and differences of platelet counts using different channels between Mindray BC-6800Plus and Sysmex XN-9000.

PLT Count * (×10^9^/L)	BC-6800P PLT-I vs. XN PLT-I			BC-6800P PLT-O vs. XN PLT-F		
	Equation	r (95% CI)	Mean Difference (95% CI)	Equation	r (95% CI)	Mean Difference (95% CI)
≤10 (*n* = 38)	y = 0.98x − 0.13	0.80 (0.64–0.89)	−0.2 (−6.2–5.8)	y = 1.14x + 0.92	0.83 (0.70–0.91)	−2.1 (−6.5–2.3)
11–20 (*n* = 112)	y = 1.00x + 0.00	0.57 (0.43–0.68)	−0.2 (−8.5–8.1)	y = 1.25x − 2.50	0.76 (0.67–0.83)	−1.8 (−6.0–2.4)
21–50 (*n* = 111)	y = 1.15x − 5.20	0.83 (0.76–0.88)	3.6 (−68.1–75.3)	y = 1.00x + 2.00	0.94 (0.91–0.96)	−1.6 (−6.9–3.8)
51–100 (*n* = 60)	y = 1.07x − 4.36	0.94 (0.89–0.96)	−0.9 (−11.8–10.1)	y = 1.04x − 1.07	0.93 (0.89–0.96)	−2.5 (−13.5–8.5)
>100 (*n* = 195)	y = 1.04x − 1.58	0.99 (0.98–0.99)	−7.7 (−30.4–15.0)	y = 0.96x + 5.24	0.99 (0.99–0.99)	3.2 (−16.5–22.9)
Total (*n* = 516)	y = 1.03x − 0.61	0.99 (0.99–0.99)	−2.3 (−39.7–35.1)	y = 0.98x + 2.62	1.00 (1.00–1.00)	0.0 (−13.9–14.0)

All *p*-values were <0.001. * PLT counts measured by XN-F were considered as reference. Abbreviations: BC-6800P, Mindray BC-6800Plus; CI, confidence interval; PLT, platelet; XN, Sysmex XN-9000.

**Table 2 diagnostics-12-00068-t002:** Agreement of platelet counts using different channels between Mindray BC-6800Plus and Sysmex XN-9000.

XN PLT-I (×10^9^/L) Total (*n* = 516)	BC-6800P PLT-I (×10^9^/L)					XN PLT-F (×10^9^/L)Total (*n* = 516)	BC-6800P PLT-O (×10^9^/L)				
	≤10 (*n* = 35)	11–20 (*n* = 81)	21–50 (*n* = 144)	51–100 (*n* = 58)	>100 (*n* = 198)		≤10 (*n* = 28)	11–20 (*n* = 93)	21–50 (*n* = 139)	51–100 (*n* = 58)	>100 (*n* = 198)
≤10 (*n* = 32)	30	2	0	0	0	≤10 (*n* = 38)	28	10	0	0	0
11–20 (*n* = 88)	5	64	19	0	0	11–20 (*n* = 112)	0	82	30	0	0
21–50 (*n* = 140)	0	15	121	4	0	21–50 (*n* = 111)	0	1	109	1	0
51–100 (*n* = 60)	0	0	3	54	3	51–100 (*n* = 60)	0	0	0	57	3
>100 (*n* = 196)	0	0	1	0	195	>100 (*n* = 195)	0	0	0	0	195
Cohen’s weighted kappa = 0.93 (0.91–0.95) (0.76 (0.69–0.83)) *						Cohen’s weighted kappa = 0.94 (0.93–0.96) (0.78 (0.72–0.84)) *					

* Cohen’s weighted kappa agreement in PLT counts ≤ 50 × 10^9^/L (*n* = 261). Abbreviations: BC-6800P, Mindray BC-6800Plus; PLT, platelet; XN, Sysmex XN-9000.

**Table 3 diagnostics-12-00068-t003:** The precision of platelet counts using different channels of Mindray BC-6800Plus and Sysmex XN-9000 in 10 thrombocytopenic samples.

	PLT-I				PLT-O (or PLT-F)			
	BC-6800P		XN		BC-6800P		XN	
	Mean (SD), ×10^9^/L	CV, %	Mean (SD), ×10^9^/L	CV, %	Mean (SD), ×10^9^/L	CV, %	Mean (SD), ×10^9^/L	CV, %
1	13.5 (1.2)	8.7	20.1 (3.3)	16.3	18.5 (0.5)	**2.8**	15.9 (1.2)	21.7
2	13.7 (2.0)	14.6	16.2 (2.3)	14.2	13.2 (0.6)	**4.8**	12.1 (1.3)	10.6
3	13.9 (1.8)	12.9	17.3 (2.8)	16.4	16.2 (0.9)	**5.7**	13.9 (1.4)	9.9
4	7.9 (2.6)	32.9	9.7 (2.7)	27.9	8.9 (0.3)	**3.6**	7.6 (0.7)	9.2
5	14.3 (1.2)	8.1	15.8 (1.5)	9.3	18.1 (0.7)	**4.1**	15.4 (0.5)	**3.4**
6	18.3 (1.3)	**7.3**	17.8 (1.7)	**9.5**	20.3 (0.5)	**2.4**	16.2 (0.6)	**3.9**
7	18.0 (1.3)	**7.4**	18.6 (1.5)	**8.1**	21.0 (0.8)	**3.9**	16.9 (1.0)	**5.9**
8	12.1 (1.7)	13.7	11.8 (3.1)	26.1	7.0 (0.7)	9.5	3.9 (0.3)	8.1
9	16.8 (2.0)	12.2	20.2 (4.5)	22.1	11.2 (0.6)	**5.6**	8.8 (0.8)	9.0
10	17.7 (2.1)	11.9	21.5 (2.2)	10.1	19.3 (0.5)	**2.5**	18.5 (0.5)	**2.8**

Bold types indicate less than desirable specification of precision (7.6 CV%) from the EFLM Biological Variation Database [25]. Abbreviations: BC-6800P, Mindray BC-6800Plus; CV%, percent coefficient of variation; PLT, platelet; SD, standard deviation; XN, Sysmex XN-9000.

**Table 4 diagnostics-12-00068-t004:** The precision of clinical reportable platelet parameters of Mindray BC-6800Plus and Sysmex XN-9000 in 10 thrombocytopenic samples.

	IPF (%)				MPV (fL)				PCT (%)				PDW (fL)				P-LCR (%)			
	BC-6800P		XN		BC-6800P		XN		BC-6800P		XN		BC-6800P		XN		BC-6800P		XN	
	Mean (SD)	CV, %	Mean (SD)	CV, %	Mean (SD)	CV, %	Mean (SD)	CV, %	Mean (SD)	CV, %	Mean (SD)	CV, %	Mean (SD)	CV, %	Mean (SD)	CV, %	Mean (SD)	CV, %	Mean (SD)	CV, %
1	7.3 (0.4)	5.5	2.7 (0.5)	20.4	10.2 (0.6)	6.2	12.5 (0.6)	4.6	0.026 (0.040)	152.092	0.03 (0.00)	13.23	15.9 (0.6)	3.6	14.1 (1.4)	9.9	29.4 (5.2)	17.7	42.5 (2.5)	5.9
2	2.6 (0.6)	21.5	0.9 (0.2)	27.7	9.4 (0.8)	8.5	12.5 (1.6)	12.5	0.013 (0.003)	20.165	0.02 (0.01)	35.36	16.5 (0.7)	4.5	13.2 (6.3)	48.1	26.5 (6.4)	24.2	42.6 (7.6)	17.8
3	8.4 (0.8)	9.2	10.9 (1.6)	12.5	10.7 (0.7)	6.4	NA	NA	0.015 (0.003)	18.298	NA	NA	16.0 (0.3)	1.7	NA	NA	40.0 (4.1)	10.3	NA	NA
4	9.3 (0.8)	8.9	11.1 (1.5)	13.1	10.2 (1.1)	11.1	NA	NA	0.009 (0.003)	33.432	NA	NA	15.8 (0.5)	3.2	NA	NA	38.1 (9.9)	25.9	NA	NA
5	1.5 (0.3)	19.2	0.4 (0.2)	46.1	8.2 (0.3)	4.2	10.2 (0.6)	6.1	0.012 (0.001)	10.248	0.02 (0.00)	28.41	15.3 (0.5)	3.2	11.3 (1.5)	13.7	13.3 (3.3)	25.0	26.6 (4.0)	14.9
6	5.1 (0.4)	7.8	5.4 (0.5)	9.3	11.0 (0.5)	4.8	12.4 (1.0)	7.9	0.020 (0.002)	9.377	0.02 (0.00)	15.79	16.8 (0.4)	2.4	15.5 (1.7)	10.7	35.4 (3.4)	9.7	43.3 (4.7)	10.8
7	4.2 (0.5)	12.4	1.5 (0.4)	25.3	10.3 (0.6)	6.0	11.8 (0.4)	3.6	0.035 (0.051)	147.762	0.02 (0.00)	15.06	16.1 (0.4)	2.6	15.3 (2.8)	18.4	28.4 (4.0)	14.0	41.1 (3.8)	9.2
8	4.2 (0.8)	18.5	5.3 (1.1)	21.3	9.9 (0.7)	7.5	NA	NA	0.012 (0.003)	23.011	NA	NA	16.4 (0.2)	1.2	NA	NA	35.9 (4.2)	11.7	NA	NA
9	2.1 (0.5)	25.5	1.6 (0.4)	23.8	10.1 (0.9)	8.8	NA	NA	0.017 (0.004)	21.785	NA	NA	16.0 (0.5)	3.2	NA	NA	34.8 (7.8)	22.4	NA	NA
10	5.4 (0.5)	9.4	6.7 (0.7)	10.4	10.1 (0.8)	7.6	11.8 (0.8)	6.4	0.022 (0.004)	16.421	0.03 (0.01)	18.75	16.9 (0.6)	3.5	14.6 (1.6)	10.7	29.8 (4.1)	13.6	38.9 (3.6)	9.4

Abbreviations: BC-6800P, Mindray BC-6800Plus; CV%, percent coefficient of variation; NA, not available; SD, standard deviation; XN, Sysmex XN-9000.

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
