# Peer review of "Performance of Platelet Counting in Thrombocytopenic Samples: Comparison between Mindray BC-6800Plus and Sysmex XN-9000"

_diagnostics, 2021, doi:10.3390/diagnostics12010068_

Round 1
Reviewer 1 Report
Dear Authors,
Overall, the article is well-written. The idea is adequate, the method is suitable. The paper has a short introduction that gives the reader enough background to understand the experimental setup and result from the analysis.
The number of samples gives a possibility to make an adequate statistical analysis. Statistical methods are accurately used and the analysis is well performed.
It is well known the advantage of the fluorescence methods vs impedance methods, especially for samples with low platelet concentrations. In my opinion, the manuscript would be improved if the discussion section would be expanded in this direction. In this regard, the reference articles should be expanded, especially since the authors (as they themselves note) did not use the reference method (namely immunological PLT counting).
After these minor corrections, I can propose the paper for publication in Diagnostics.
Author Response
Response to reviewer’s comments
diagnostics-1512855-R
Performance of platelet counting in thrombocytopenic samples: comparison between Mindray BC-6800Plus and Sysmex XN-9000
We really appreciate both of you for your time and effort on our manuscript. During the revision, we recognized the weakness of the manuscript and could improve its quality. We agree with the issues that you pointed out and tried to reflect them in the revised manuscript. We hope our effort would be satisfactory to you.
Reviewer #1:
- It is well known the advantage of the fluorescence methods vs impedance methods, especially for samples with low platelet concentrations. In this regard, the reference articles should be expanded, especially since the authors (as they themselves note) did not use the reference method (namely immunological PLT counting).
According to the comment, we added the following sentences in the discussion section. We also added two references.
It is well known that the fluorescence methods have more advantages over impedance methods, especially for samples with low PLT counts [11, 12, 20, 21, 27]. Our data support the previous findings. (Page 9, line 195)
- Guo, P.; Cai, Q.; Mao, M.; Lin, H.; Chen, L.; Wu, F.; Wang, J. Performance evaluation of the new platelet measurement channel on the BC-6800 Plus automated hematology analyzer. Int J Lab Hematol 2021, ahead of print; doi: 10.1111/ijlh.13753.
- Michelson, A.; Cattaneo, M.; Frelinger, A.; Newman, P. ed. Platelets. 4th ed.; Elsevier Science: 2019; pp. 581-591. ISBN: 978-0-12-813456-6.
Reviewer 2 Report
Regarding to the manuscript ID: diagnostics-1512855-peer-review-v1 entitled The performance of platelet (PLT) counting….. authored by Kim et al., . The work is interesting, valuable and add significant data in the field of hematology. We thank the authors for this idea, we accepted the paper with minor revision. however, some comments should be addressed in this manuscript.
Why authors selected both types of Mindray BC- 9 6800Plus and Sysmex XN-9000 in their study?
The discussion needs further clarifications, it’s not enough.
Pleases, adjust the references according to the journal guidelines?
Specific comments
Line 16 …. XN PLT-F was arranged between high to very high.
Line 17 . please use other word more accurate than ……..very good agreement.
Live 46 …….. CV please insert the full name for the first observation.
Line 47 …...ransfusion thresholds [14-16]. add full stop.
Line 74 …PLT counts measured by XN PLT-F were considered as reference. Please improve this sentence.
Line 200 please, rewrite this sentence, it is unclear.
Line 204 However, BC-6800P PLT-I and XN PLT-I did not. May be needed to revise.
Line 208 5.6 CV% was a median, please start the sentence with letter A 5.6% CV ……. .
Line 2019 in thrombocytopenic samples (≤ 100 x 109/L) (Table 1, Table 2, Table 3), you can be concise to (Table 1-3).
Author Response
Response to reviewer’s comments
diagnostics-1512855-R
Performance of platelet counting in thrombocytopenic samples: comparison between Mindray BC-6800Plus and Sysmex XN-9000
We really appreciate both of you for your time and effort on our manuscript. During the revision, we recognized the weakness of the manuscript and could improve its quality. We agree with the issues that you pointed out and tried to reflect them in the revised manuscript. We hope our effort would be satisfactory to you.
Reviewer #2:
- Why authors selected both types of Mindray BC-6800Plus and Sysmex XN-9000 in their study?
According to the comment, we added the following sentences in the introduction section. We also added a reference.
Differently from BC-6800, BC-6800P measures PLT using multi-fold counting (x 8 times) automatically in thrombocytopenic samples [19, 20]. BC-6800P showed acceptable precision using six thrombocytopenic samples (< 20 x 109/L) [20]. (page 2, line 52)
We evaluated the performance of PLT counting of BC-6800P compared with XN, which is currently used as a routine hematology analyzer in our laboratory, according to their PLT measurement methods. (page 2, line 62)
- Guo, P.; Cai, Q.; Mao, M.; Lin, H.; Chen, L.; Wu, F.; Wang, J. Performance evaluation of the new platelet measurement channel on the BC-6800 Plus automated hematology analyzer. Int J Lab Hematol 2021, ahead of print; doi: 10.1111/ijlh.13753.
- The discussion needs further clarifications, it’s not enough.
According to the comment, we added the following sentences in the discussion section. We also added references.
It is well known that the fluorescence methods have more advantages over impedance methods, especially for samples with low PLT counts [11, 12, 20, 21, 26]. Our data support the previous findings. (page 9, line 195)
Clinically reportable PLT parameters, including IPF, are useful for diagnosing and monitoring thrombocytopenia. Based on the present and previous findings, the PLT counting methods also seem to affect precision and reference intervals of clinically reportable PLT parameters (Table 4) [19, 21, 29-31]. Thus, further exploration on clinically reportable PLT parameters across PLT counting methods is necessary to promote their clinical utilization. (page 9, line 216)
- Guo, P.; Cai, Q.; Mao, M.; Lin, H.; Chen, L.; Wu, F.; Wang, J. Performance evaluation of the new platelet measurement channel on the BC-6800 Plus automated hematology analyzer. Int J Lab Hematol 2021, ahead of print; doi: 10.1111/ijlh.13753.
- Michelson, A.; Cattaneo, M.; Frelinger, A.; Newman, P. ed. Platelets. 4th ed.; Elsevier Science: 2019; pp. 581-591. ISBN: 978-0-12-813456-6.
- Hoffmann, J.J. Reticulated platelets: analytical aspects and clinical utility. Clin Chem Lab Med 2014, 52, 1107-1117.
- Ko, Y.J.; Hur, M.; Kim, H.; Choi, S.G.; Moon, H.W.; Yun, Y.M. Reference interval for immature platelet fraction on Sysmex XN hematology analyzer: a comparison study with Sysmex XE-2100. Clin Chem Lab Med 2015, 53, 1091-1097.
- Buttarello, M.; Mezzapelle, G.; Freguglia, F.; Plebani, M. Reticulated platelets and immature platelet fraction: Clinical applications and method limitations. Int J Lab Hematol 2020, 42, 363-370.
3.Pleases, adjust the references according to the journal guidelines?
Thank you for the comment. We checked the references again carefully.
- Gottschall, J.; Wu, Y.; Triulzi, D.; Kleinman, S.; Strauss, R.; Zimrin, A.B.; McClure, C.; Tan, S.; Bialkowski, W.; Murphyet, E.; et al. The epidemiology of platelet transfusions: an analysis of platelet use at 12 US hospitals. Transfusion 2020, 60, 46-53.
- de Bruin, S.; Scheeren, T.W.L.; Bakker, J.; van Bruggen, R.; Vlaar, A.P.J.; Cardiovascular Dynamics Section and Transfusion Guideline Task Force of the ESICM. Transfusion practice in the non-bleeding critically ill: an international online survey-the TRACE survey. Crit Care 2019, 23, 309:1-309:8.
- Borge, D.; Marcus, L. ed. A compendium of transfusion practice guidelines. 4th ed.; American Red Cross: 2021. Available online: https://www.redcrossblood.org/content/dam/redcrossblood/hospital-page-documents/334401_compendium_v04jan2021_bookmarkedworking_rwv01.pdf. (accessed on 22 December 2021).
- Harrison, P.; Ault, K.A.; Chapman, S.; Charie, L.; Davis, B.; Fujimoto, K.; Houwen, B.; Kunicka, J.; Lacombe, F.; Machinet, S.; et al. An interlaboratory study of a candidate reference method for platelet counting. Am J Clin Pathol 2001, 115, 448-459.
- Sun, Y.; Hu, Z.; Huang, Z.; Chen, H.; Qin, S.; Jianing, Z.; Chen, S.; Qin, X.; Ye, Y.; Wang, C. Compare the accuracy and preci-sion of Coulter LH780, Mindray BC-6000 Plus, and Sysmex XN-9000 with the international reference flow cytometric method in platelet counting. PLoS One 2019, 14, e0217298:1- e0217298:11.
- CoÅŸkun, A.; Carobene, A.; Kilercik, M.; Serteser, M.; Sandberg, S.; Aarsand, A.K.; Fernandez-Calle, P.; Jonker, N.; Bartlett, W.A.; Díaz-Garzónet, J.; et al. Within-subject and between-subject biological variation estimates of 21 hematological parameters in 30 healthy subjects. Clin Chem Lab Med2018, 56, 1309-1318.
- European Federation of Clinical Chemistry and Laboratory Medicine. EFLM Biological Variation Database. Available online: https://biologicalvariation.eu/search?q=thrombocytes (accessed on 21 December 2021).
- Baccini, V.; Geneviève, F.; Jacqmin, H.; Chatelain, B.; Girard, S.; Wuilleme, S.; Vedrenne, A.; Guiheneuf, E.; Tous-saint-Hacquard, M.; Everaere, F.; et al. Platelet Counting: Ugly Traps and Good Advice. Proposals from the French-Speaking Cellular Hematology Group (GFHC). J Clin Med 2020, 9, 808:1-808:27.
- CLSI. Evaluation of Precision of Quantitative Measurement Procedures, 3rd ed. CLSI document EP05-A3; Clinical and Laboratory Standards Institute: Wayne, PA, USA, 2014. ISBN 1-56238-968-8
- Line 16: XN PLT-F was arranged between high to very high.
According to the comment, we modified the following sentence in the Abstract.
The correlation of BC-6800P PLT-I and XN PLT-I was arranged moderate to very high; but the correlation of BC-6800P PLT-O and XN PLT-F was arranged high to very high. (page 1, line 16)
- Line 17: please use other word more accurate than very good agreement.
We used the term “very good agreement” according to the description in the “statistical analysis (page 3, line 123)”. To clarify it, we added the kappa values to the sentence in the Abstract.
Both BC-6800P PLT-I vs. XN PLT-I and BC-6800P PLT-O vs. XN PLT-F showed very good agreement (κ = 0.93 and κ = 0.94). (page 1, line 17)
- Line 46: please insert the full name for the first observation, CV.
Line 47: transfusion thresholds [14-16]. add full stop.
According to the comment, we modified the following sentence.
The UK National External Quality Assessment Scheme for General Haematology reported that the imprecision increased at lower levels of PLT count, showing a mean coefficient of variance (CV) as high as 15 - 32% at transfusion thresholds [14-16]. (page 2, line 47)
- Line 74: PLT counts measured by XN PLT-F were considered as reference. Please improve this sentence.
According to the comment, we modified the following sentence.
The peripheral blood samples (3mL) were collected directly into VACUETTE EDTA K3 tubes (Greiner Bio-One, Kremsmünster, Austria), and CBC was analyzed within two hours using both BC-6800P and XN. PLT counts measured by XN PLT-F were considered a reference method, and PLT counts measured by BC-6800P were considered a comparative method. (page 2, line 79)
- Line 200: please, rewrite this sentence, it is unclear.
According to the comment, we modified the following sentence.
The fluorescence methods of both analyzers showed better reproducibility than impedance methods. In addition, BC-6800P PLT-O showed better reproducibility than XN PLT-F. (page 9, line 207)
- Line 204: However, BC-6800P PLT-I and XN PLT-I did not. May be needed to revise.
According to the comment, we modified the following sentence.
However, BC-6800P PLT-I and XN PLT-I did not show the acceptable precision. (page 9, line 213)
- Line 208: 5.6 CV% was a median, please start the sentence with letter A 5.6% CV
According to the comment, we modified the following sentence.
A 7.6% CV was a median CV estimate of the within-subject of EFLM and considered a desirable PLT specification. (page 9, line 223)
- Line 219: in thrombocytopenic samples (≤ 100 x 109/L) (Table 1, Table 2, Table 3), you can be concise to (Table 1-3).
According to the comment, we modified the following sentence.
BC-6800P PLT-O enumerated PLT counts higher and more precisely than XN PLT-F in thrombocytopenic samples (≤ 100 x 109/L) (Table 1-3 and